Identification and analysis of Chrysanthemum nankingense NAC transcription factors and an expression analysis of OsNAC7 subfamily members

Wang Hai 1 2 3
http://orcid.org/0000-0002-4646-9811 Li Tong 1 2 3
http://orcid.org/0000-0002-3886-933X Li Wei 4
Wang Wang 1 2 3
Zhao Huien 1 2 3 zhaohuien@bjfu.edu.cn
1 Beijing Key Laboratory of Ornamental Plants Germplasm Innovation & Molecular Breeding, National Engineering Research Center for Floriculture , Beijing , China
2 Key Laboratory of Genetics and Breeding in Forest Trees and Ornamental Plants of Ministry of Education , Beijing , China
3 College of Landscape Architecture, Beijing Forestry University , Beijing , China
4 College of Landscape Architecture and Forestry, Qingdao Agricultural University , Qingdao, Shandong , China
Ma Wei
Electronic publication date: 2021 May 26
Publication date: 2021
Volume: 9
Electronic Location ID: e11505
Received 2021 Feb 5; Accepted 2021 May 3
Copyright: © 2021 Wang et al.
Copyright year: 2021
Copyright holder: Wang et al.
License: This is an open access article distributed under the terms of the Creative Commons Attribution License, which permits unrestricted use, distribution, reproduction and adaptation in any medium and for any purpose provided that it is properly attributed. For attribution, the original author(s), title, publication source (PeerJ) and either DOI or URL of the article must be cited.
License URL: https://creativecommons.org/licenses/by/4.0/

Keywords: Chrysanthemum nankingense, CnNAC TFs, OsNAC7, Osmotic stress, Salt stress, Growth and development

Funding: National Key R&D Program of China 2018YFD1000401 Beijing Forestry University 2019XKJS0323 This work was supported by the National Key R&D Program of China (2018YFD1000401) and the World-Class Discipline Construction and Characteristic Development Guidance Funds for Beijing Forestry University (2019XKJS0323). The funders had no role in study design, data collection and analysis, decision to publish, or preparation of the manuscript.

==============================
NAC (NAM, ATAF1-2, and CUC2) transcription factors (TFs) play a vital role in plant growth and development, as well as in plant response to biotic and abiotic stressors (Duan et al., 2019; Guerin et al., 2019). Chrysanthemum is a plant with strong stress resistance and adaptability; therefore, a systematic study of NAC TFs in chrysanthemum is of great significance for plant breeding. In this study, 153 putative NAC TFs were identified based on the Chrysanthemum nankingense genome. According to the NAC family in Arabidopsis and rice, a rootless phylogenetic tree was constructed, in which the 153 CnNAC TFs were divided into two groups and 19 subfamilies. Moreover, the expression levels of 12 CnNAC TFs belonging to the OsNAC7 subfamily were analyzed in C. nankingense under osmotic and salt stresses, and different tissues were tested during different growth periods. The results showed that these 12 OsNAC7 subfamily members were involved in the regulation of root and stem growth, as well as in the regulation of drought and salt stresses. Finally, we investigated the function of the CHR00069684 gene, and the results showed that CHR00069684 could confer improved salt and low temperature resistance, enhance ABA sensitivity, and lead to early flowering in tobacco. It was proved that members of the OsNAC7 subfamily have dual functions including the regulation of resistance and the mediation of plant growth and development. This study provides comprehensive information on analyzing the function of CnNAC TFs, and also reveals the important role of OsNAC7 subfamily genes in response to abiotic stress and the regulation of plant growth. These results provide new ideas for plant breeding to control stress resistance and growth simultaneously.

Introduction

Transcription factors (TFs) are a class of important regulatory proteins, which inhibit or enhance gene expression by covalently binding to the DNA binding domain (Fang et al., 2008). NAC (NAM, ATAF1-2, and CUC2) TFs are a large family endemic to plants and are rarely found in the genomes of bacteria, fungi, or animals (Nuruzzaman et al., 2010). NAC family proteins share a common NAC domain, which is identified by a consistent NAM sequence from Petunia and ATAF1/ATAF2 and CUC2 sequences from Arabidopsis. Arabidopsis ATAF is the first described member of NAC. NAM (NO APICAL MERISTEM) plays an important role determining the apical meristem and the location of primordia. Petunias with a NAM gene mutation cannot form apical meristem (Souer et al., 1996). The CUC2 gene plays a role in the development of the embryo and flower, and mutations in CUC1 and CUC2 cause the separation of cotyledons (embryonic organs), sepals, and stamens (floral organs), leading to defects in the formation of terminal bud meristematic tissues (Aida et al., 1997; Aida, Ishida & Tasaka, 1999; Ishida et al., 2000). NAC family genes have about 160 residues at the N-terminus, forming a typical highly conserved domain, which is divided into five subdomains. Among them, three subdomains are highly conserved, and two subdomains are diverse. These two subdomains are related to the multi-functions of the NAC family TFs (Duan et al., 2019; Ooka et al., 2003).

NAC TFs are expressed during different development stages and in different tissues of plants, which are closely related to plant growth and development, as well as the regulation of plant response to abiotic and biotic stresses (Liu et al., 2018). Ooka et al. (2003) comprehensively analyzed 75 predicted NAC TFs in rice (Oryza sativa) and 105 predicted NAC TFs in Arabidopsis (Arabidopsis thaliana) and divided the NAC TFs into two groups (Group 1 and Group 2) according to similarities in their structural domains. Group 1 contains 14 subfamilies and Group 2 contains four subfamilies. Rushton et al. (2008) analyzed the evolutionary relationships among 450 NAC proteins in rice (O. sativa), Arabidopsis (A. thaliana), tobacco (Nicotiana tabacum), poplar (Populus trichocarpa), and other plants, and divided the NAC gene family into seven subfamilies. Among them, six subfamilies exist in all plants, while the other subfamily only occurs in tobacco, pepper, potato, and tomato. So far, 151 NAC family members have been found in O. sativa (Nuruzzaman et al., 2010), 117 in A. thaliana (Nuruzzaman et al., 2010; Ooka et al., 2003), 152 in tobacco (N. tabacum) (Rushton et al., 2008), 205 in soybean (Glycine max) (Le et al., 2011), 104 in tomato (Solanum lycopersicum) (Su et al., 2015), 163 in poplar (P. trichocarpa) (Hu et al., 2010), 79 in grape (Vitis vinifera) (Ju et al., 2020), 168 in durum wheat (Triticum turgidum) (Saidi, Mergby & Brini, 2017), and 488 in bread wheat (Triticum aestivum) (Guerin et al., 2019).

Studies on NAC TFs have reported that the NAC genes respond to at least one biotic or abiotic stress, such as saline-alkali, drought, cold, or exogenous abscisic acid (ABA), which play important roles in plant drought tolerance, saline-alkali tolerance, virus resistance, and the cold resistance response (Ju et al., 2020; Gao et al., 2018). In previous studies, most NAC TFs are closely related to salt and alkali tolerance, cold tolerance, and disease resistance of plants, whereas other NAC TFs improve plant resistance to multiple stressors at the same time (Sun et al., 2019; Puranik et al., 2012; Xia et al., 2010). The A. thaliana NAC TFs ATAF1, ANAC016, ANAC019, ANAC055, and ANAC072 (Tran et al., 2004; Wu et al., 2009); the O. sativa NAC TFs OsNAC045, OsNAC10, and OsNAC022 (Jeong et al., 2010); the G. max NAC TFs GmNAC085, GmNAC109, GmNAC30, GmNAC81, GmNAC003, GmNAC004, GmNAC11, and GmNAC20 (Nguyen et al., 2018; Yang et al., 2019; Hao et al., 2011); and the T. turgidum NAC TF TaRNAC1 (Hao et al., 2011; Chen et al., 2018) enhance drought tolerance. The Lilium regale NAC TF LrNAC35, the S. lycopersicum NAC TFs SlNAC1 and SlNAC61, and the A. thaliana NAC TFs ATAF2 and TIP are all related to virus resistance in plants (Sun et al., 2019; Huang et al., 2017; Selth et al., 2005; Wang, Goregaoker & Culver, 2009; Donze et al., 2014). The G. max NAC TF GmNAC20 significantly improves cold resistance in plants (Puranik et al., 2012; Hao et al., 2011; Liu et al., 2012). The O. sativa NAC TFs OsNAC5 and OsNAC6 (the ATAF subfamily) improve drought resistance, salt tolerance, and freeze resistance of transgenic plants (Nakashima et al., 2007; Song et al., 2011) and the V. vinifera NAC TF VvNAC17 enhances the sensitivity of A. thaliana to ABA and improves salt tolerance, cold tolerance, and drought tolerance (Ju et al., 2020).

Most genera in Anthemideae are distributed in southern Africa and the Mediterranean region, but are also widely distributed in Europe and Asia. The main economic crops in the family include Chrysanthemum carinatum Schousb., C. segetum L., C. nankingense, Dendranthema morifolium (Ramat.) Tzvel., Pyrethrum cinerariifolium Trev., and Artemisia annua. These are important vegetables, ornamental flowers, grasses, and herbs. Although Anthemideae plants are widely distributed, only some of the currently available plants can be cultivated in arid, saline, or cold regions, as their various traits are weakened and can only be planted as single-season cultivars. Under adverse circumstances, they lose the advantages of being an easy-to-manage perennial, leading to difficulties in popularity and application. Improving the growth characteristics and resistance to drought and saline-alkali stresses of Anthemideae plants would significantly improve their application range and enhance their cultural advantages. NAC TFs play an important role in abiotic stress in plants (Puranik et al., 2012). Therefore, regulating the CnNAC TFs is an effective way to enhance the resistance of Anthemideae plants. However, most studies on the Chrysanthemum NAC genes are single-gene studies that focus on disease and insect resistance. Few comprehensive studies have been performed on CnNAC TFs and their abiotic stress responses. In this study, CnNAC TFs were analyzed based on published C. nankingense genome data. In addition, the expression levels of the CnNAC TFs belonging to the OsNAC7 subfamily were analyzed in different tissues at different growth stages in C. nankingense under osmotic and salt stresses. These results are expected to contribute to resistance breeding and growth and developmental regulation of Chrysanthemum.

Materials & Methods

Gene retrieval of C. nankingense NAC TFs

The A. thaliana NAC TF sequences were obtained from the Arabidopsis Information Resource (http://www.arabidopsis.org/). C. nankingense genomic data were downloaded from the website of the Institute of Chinese Medicine, Chinese Academy of Chinese Medical Sciences (http://www.amwayabrc.com/zh-cn/download.htm). Based on the A. thaliana NAC protein sequences, HMMER3.0 software was used to construct a multiple comparison HMM model, which was then used to search the C. nankingense genome protein data to obtain the CnNAC protein sequences. The CnNAC protein sequences were identified and screened using the SMART tool (http://smart.embl-heidelberg.de/) and Pfam (https://pfam.xfam.org/), and the predicted CnNAC TF members were obtained. The CnNAC TFs were used for subsequent analyses based on this comprehensive screening (Li et al., 2019).

Bioinformatics analysis of C. nankingense NAC TFs

The NAC TFs of C. nankingense, A. thaliana, and some of those of O. sativa were used to construct a rootless phylogenetic tree using the multiple comparison function in ClustalW (default parameters) and the neighbor-joining (NJ) method in MEGA-X software according to the existing A. thaliana and O. sativa NAC gene family classification system (Duan et al., 2019; Ooka et al., 2003; Hu et al., 2015). The rootless phylogenetic tree was used to study the phylogenetic relationships of the CnNAC TFs (Zhang et al., 2019). MEME online software (http://meme-suite.org/tools/meme) was used to predict the conserved motifs of the CnNAC TFs. The ProtParam online tool (https://web.expasy.org/protparam/) was used to analyze the physical and chemical properties of the CnNAC TF amino acid sequences and predict their subcellular locations (Li et al., 2019). The TMHMM online tool (http://www.cbs.dtu.dk/services/TMHMM-2.0/) was used to predict the protein transmembrane structure of the CnNAC TFs. The three-dimension (3D) models of the conserved NAC domains of the CnNAC proteins were predicted by Phyre2 (http://www.sbg.bio.ic.ac.uk/phyre2/html/page.cgi?id=index) (Kelley et al., 2015).

Growth conditions and treatment of the plant materials

C. nankingense seeds (from Beijing Forestry University) were sown on peat to produce seedlings under growing conditions of 25 °C/23 °C (day/night), 16 h/8 h (day/night), and 2,000 Lux. The roots, stems, and 3–5 new leaves on the top of the seedlings were collected and quickly flash frozen in liquid nitrogen as experimental materials at 28, 44, and 60 d after seeding, respectively, and stored at −80 °C. Three replicates were taken for each period.

Rooted C. nankingense seedlings obtained from stem hydroponics were subjected to osmotic and salt stresses. The side shoots of C. nankingense seedlings grown for 44 d were cut out and inserted into cultivation medium containing MS medium and 0.1 mg/L IBA. The growing conditions were 25 °C/23 °C (day/night), 16 h/8 h (day/night), and 2,000 Lux. The cultivation medium was changed every 3–5 d, and rooted seedlings were obtained after 20 d. Then, the cultivation solution was changed to 20% PEG6000 or 200 mM NaCl solution for the osmotic and salt stress treatments, respectively. Samples were taken at 0, 1, 4, 8, 12, 24, and 36 h (Fan et al., 2016). The entire rooted seedling was taken as a sample and stored at −80 °C after flash-freezing in liquid nitrogen. Three replicates were taken for each period.

RNA isolation and real-time quantitative-polymerase chain reaction (RT-qPCR) analysis

Total RNA was isolated using the FastPure Plant Total RNA Isolation Kit (Code No. RC401-01; Nanjing Vazyme Biotechnical Co., Ltd., Nanjing, China). The reverse transcription of RNA was carried out using the PrimeScript™ RT reagent Kit with the gDNA Eraser (Perfect Real Time, Code No. RR047A; Takara Bio., Shiga, Japan). The primers used to analyze the expression of the OsNAC7 subfamily members were designed based on the C. nankingense genome (Table 1). RT-qPCR was carried out with on the Applied Biosystems StepONEPlus Real Time PCR System (Foster City, CA, USA), using ChamQ Universal SYBR qPCR Master Mix (Code No. Q711-03; Nanjing Vazyme Biotechnical Co., Ltd., Nanjing, China). The reaction system was 40 cycles of 95 °C for 20 s, 95 °C for 3 s, 60 °C for 30 s, 95 °C for 15 s, and 60 °C for 1 min (Wei et al., 2016), with actin7 from C. nankingense as the internal reference gene (Chen et al., 2018). The relative expression of genes was calculated by the 2−ΔCt method.

Table 1 RT-qPCR primers for the OsNAC7 subfamily genes of C. nankingense.

Number	Gene name	Primer sequence	GC CONTENT(%)	TM (°C)	
1	actin7-F	CGTTGCCCTGAGGTTCTCTT	55	60.0	
actin7-R	CCTTGCTCATCCTGTCAGCA	55	60.0	
2	CHR00069684-F	CCAAGGCTCGATAGTCCCAC	60	59.9	
CHR00069684-R	AGGAAGTCAGTTGCATGGGG	55	60	
3	CHR00042500-F	GAGCACCTCATGGCCAGAAA	55	60.3	
CHR00042500-R	CACGACATACCACCCAACCA	55	60	
4	CHR00020838-F	AAAGGACGAGCACCTCATGG	55	60	
CHR00020838-R	ACCCACCCCTCTTCTTGAGT	55	60.1	
5	CHR00009966-F	CACCTCCCACGGCTTGATAA	55	59.7	
CHR00009966-R	GTCAGCAAGTCACTCACGGA	55	60	
6	CHR00048081-F	ACTCCTGAAGATGGGTGGGT	55	60.2	
CHR00048081-R	GCCTGCGAAAAAGAACCAGC	55	60.7	
7	CHR00043953-F	ATCTACCCGCCGAAGAGTCA	55	60.4	
CHR00043953-R	GAGCTGCTTCAGAGGTGTCA	55	59.7	
8	CHR00073261-F	CCGACGAGGAACTGCTTCAT	55	60.1	
CHR00073261-R	GGCGTCGACCCTATCTTACA	55	59	
9	CHR00032377-F	GGAAGGGACAAGGCAGTGTT	55	60.2	
CHR00032377-R	CATGATGGGACGACTGGAGG	60	59.9	
10	CHR00026420-F	GTGGTGTGTCGTGCATTCAA	55	59.3	
CHR00026420-R	ATTTACCTCTGTGGCTGGGC	55	60	
11	CHR00003673-F	ACGGGGAGAGATAAGGCAGT	55	60	
CHR00003673-R	TCGGTCTTCTGTCCGTTTGG	55	60	
12	CHR00027311-F	AAGAGTTGTGCAGGCTTGGT	55	60.1	
CHR00027311-R	GCTTTCCAAAATCCAGCTGCA	47.6	60	
13	CHR00043175-F	GCACCAACAGGGCTACAAAG	55	59.4	
CHR00043175-R	CGTTAGGAGCTCGACCCTTG	60	60.2	
Note:

actin7-F/R primer is an internal reference gene primer.

Subcellular localization and functional analysis

The super1300 vector was used as the overexpression vector (kindly provided by Professor Huitang Pan of Beijing Forestry University, Beijing, China. Data S1). The vector uses super35S as the promoter. According to the open reading frame (ORF) of the CHR00069684 gene (Data S2) and the polyclonal sites of the super1300 vector, the endonucleases PstI and KpnI (NEB, Beijing, China) were selected to construct the binary expression vector according to the double enzyme method. The recombinant plasmid was transformed into Escherichia coli competent cells (DH5α), which were inoculated on solid LB medium containing 50 mg/L kanamycin; the resistance E. coli cells were validated by colony PCR. The obtained positive plaques were sent to the sequencing company (Beijing Ruibio BiotechCo.,Ltd, Amsterdam, the Netherlands) for sequencing and plasmid recovery. The super35S::GFP and super35S::CHR00069684::GFP vectors were introduced into Agrobacterium tumefaciens strain GV3101 (Weidi Biotechnology, Shanghai, China) by the freeze-thaw method and then inoculated on LB medium supplemented with 50 mg/L kanamycin, 50 mg/L gentamicin, and 50 mg/L rifampicin. The surviving plaques were validated by colony PCR, and the positive plaques were inoculated into LB liquid medium containing 50 mg/L kanamycin, 50 mg/L gentamicin, and 50 mg/L rifampicin for propagation. The transgenic tobacco (NT78) was obtained through A. tumefaciens-mediated transformation (Horsch et al., 1985), screened by 30 mg/L hygromycin, and validated by PCR identification of the target gene. CHR000696884::GFP fusion protein and GFP protein signals were observed by a laser confocal microscopy (Leica TCS SP5) in roots, stems, and leaves of the transgenic tobacco plants. The super35S::GFP vector was used as control.

Results

Identification of the CnNAC TFs and the phylogenetic analysis

According to the A. thaliana NAC protein sequences, 153 predicted CnNAC TFs were identified from the C. nankingense genome data using HMMER3.0. Using ClustalW in MEGA, 153 C. nankingense NAC proteins, 105 A. thaliana NAC proteins, and 6 O. sativa NAC proteins were multi-sequence aligned, and a rootless phylogenetic tree was constructed using the NJ method by MEGA (Fig. 1, Data S3). According to the existing A. thaliana and O. sativa NAC gene family classification system (Duan et al., 2019; Ooka et al., 2003; Hu et al., 2015) and the topology of the rootless phylogenetic tree, the CnNAC TFs were divided into two groups of 19 subfamilies, including 17 known subfamilies and two unknown subfamilies. Group 1 contained 15 subfamilies, including OsNAC7, NAC1, NAM, ANAC077, ANAC011, OsNAC8, TIP, NAC2, ONAC022, TERN, SENU5, NAP, AtNAC3, ATAF, and unclassified 2, with a total of 111 members. Group 2 contained five subfamilies, which were ANAC063, ONAC003, ANAC007, unclassified 1, and ONAC001, with a total of 42 members. Among them, the ANAC077 subfamily occurred in both groups. The different branches of the phylogenetic tree testify to the functional diversity of this family.

Figure 1 Phylogenetic tree analysis of NAC transcription factors from A. thaliana, O. sativa, and C. nankingense.

The phylogenetic trees were derived using the neighbor-joining (NJ) method by 1,000 bootstrap replications. The black circles on the outer ring represent Group1 and Group2. Each color inside represents a subfamily. The red dots represent C. nankingense. The blue dots represent rice. The unmarked represents Arabidopsis thaliana. Numbers on the branches indicate differences between different genes, and higher numbers indicate higher relatedness.

According to further analysis of the CnNAC TFs (Fig. S1), 104 NAC genes were detected in the 17 known subfamilies, and 49 NAC genes were detected in the two unknown subfamilies. Three known subfamilies (ONAC022, NAC2, and OsNAC7) contained 13, 11, and 12 members respectively, with the greatest numbers of genes among all known subfamilies. However, the two unknown subfamilies, unclassified 1 and unclassified 2, contained 28 and 21 NAC genes, respectively.

Prediction of the protein structure is considered as a reliable analytical method to understand the molecular function of proteins (Kelley et al., 2015). Therefore, the 3D models of one gene from each branch of the NAC phylogenetic trees (Fig. 1), and all 60 CnNAC family members were constructed using the Phyre2 server (Fig. 2). c3ulxA (Chen et al., 2011) and d1ut7a (Ernst et al., 2004) were used as the reference model template, and the confidence level was set as 100%. The c3ulxA template is a DNA binding protein and annotated as stress-induced NAC1 according to the crystal structure of the conserved domain of rice stress responsive NAC1 in the PDB database. The d1ut7a template is annotated as a NAC domain. The 3D modeling results revealed that these CnNAC TFs possessed similar tertiary structures, implying that CnNAC TFs may evolved from the same ancestor sequence and/or under purifying selection to maintain stabilization during the long-term evolution after the initial divergence (Zhu et al., 2019).

Figure 2 Predicted three-dimensional structure of NAC TFs from C. nankingense.

(A) Protein structure of different subfamilies in Group1. (B) Protein structure of different subfamilies in Group2. One gene from each branch of NAC phylogenetic trees is selected for prediction (Fig. 1). All genes of the OsNAC7 subfamily were predicted and compared with members of the OsNAC7 subfamily of A. thaliana NAC TFs.

Analysis of the CnNAC TF conserved motifs

The online tool MEME-Motif Suite was used to analyze the motifs of the 153 predicted CnNAC TFs (Fig. 3) and the positions of the 15 motifs on different genes were determined (Fig. 4). As a result (Fig. S2), five genes (CHR00045488-RA, CHR00007510-RA, CHR00036356-RA, CHR00079376-RA, and CHR00089260-RA) containing 11 motifs, respectively, constituted the most abundant motifs. These five genes all belonged to the unknown unclassified 1 subfamily. Nine genes (CHR00007511-RA, CHR00035414-RA, CHR00039887-RA, CHR00039888-RA, CHR00055754-RA, CHR00055752-RA, CHR00088453-RA, CHR00024060-RA, and CHR00058504-RA) containing 10 motifs, respectively, were also in the unknown subfamilies (six in unclassified 1 and three in unclassified 2). Sixty-two genes, which contained the highest number of 7 motifs, were distributed in 12 known subfamilies and two unknown subfamilies.

Figure 3 Sequence logos of conserved domains in C. nankingense.

The overall height of the stack represents the level of sequence conservation. Heights of residues within a stack indicate the frequency of each residue at that site. E-value (Expect value) is an estimate of the expected number of motifs. Sites are the number of sites contributing to the construction of the motif. Width shows the motif, each motif describes a pattern of a fixed width.

Figure 4 Conserved motifs analyses of NAC TFs in C. nankingense.

Different colors represent different motifs. P-value is the probability value test. Motif Locations shows the location of motif sites.

According to the comparative analysis of the motifs in different subfamilies (Fig. 4), the included motif bases were in motifs 1–7 in the known subfamilies of Group 1, but different subfamilies contained different numbers of motifs and different conserved motifs. For example, motifs 4, 6, 3, and 7 were in the NAM subfamily and motifs 1, 4, 2, 5, and 6 were in the OsNAC7 subfamily. However, some members in the unknown subfamily unclassified 2 had more motifs, such as motifs 10, 13, and 14, than in the other subfamilies. Each member in the known subfamilies of Group 2 contained far fewer motifs than those in the known subfamilies of Group 1. Members in the known subfamilies of Group 2 contained motifs 1, 3, 4, 5, 6, and 7, and lacked motif 2 compared to those in the known subfamilies of Group 1. However, the CnNAC TFs in unclassified 1 of Group 2 had the highest number of motifs, most of which were motifs 7–11.

Subcellular localization, protein transmembrane analysis, and physicochemical property analysis of CnNAC TFs

According to the subcellular localization and transmembrane analysis results (Table S1), 106 of the 153 CnNAC TFs were located in the nucleus and 47 were located outside the cell. Of the 111 members in Group 1, 106 were nuclear and 15 were extracellular, of which the 21 members within the unclassified 2 subfamily were all nuclear. Only 10 of the 42 members of Group 2 were located in the nucleus, the other 32 were located outside the cell, and 23 of the 28 members of the unclassified 1 subfamily of Group 2 were located outside the cell. However, the protein transmembrane analysis of the 153 NAC TFs revealed that only one member of Group 1 was expressed inside the membrane and eight members were expressed transmembrane, while the remaining 102 members were all expressed outside the membrane. There were no transmembrane-expressed Group 2 members, while two members were expressed inside the membrane and 40 members were expressed outside the membrane. The three genes expressed inside the membrane were all located in the nucleus, and the genes expressed transmembrane and outside the membrane all had members located in the nucleus and extracellularly, respectively.

The physicochemical properties of the CnNAC TFs were analyzed (Table S2). The lengths of the CnNAC TF amino acids ranged from 68 to 693 (mean length = 307.99). The molecular weight range of the CnNAC TFs was 7,787.81–80,985.3 Da with an average molecular weight of 35,195.1 Da. The theoretical isoelectric point (pI) range of the CnNAC TFs was 4.43–10.2, and the average pI was 7.09. The grand average of hydropathy was −0.72896, while the lowest value was −1.37, and the highest value was −0.275. The aliphatic amino acid index was 44.15–94.84, with a mean value of 64.58.

Analysis of the OsNAC7 subfamily gene expression levels during different C. nankingense growth stages

The OsNAC7 subfamily included 12 genes in A. thaliana, including SND1, NST1, URP7, BRn1/2, and VND1/2/3/4/5/6/7, the functions of which are all related to regulation of the formation of secondary cell walls in plant stems, roots, and anthers. Here, we analyzed the expression levels of the OsNAC7 subfamily genes in the roots, stems, and leaves of C. nankingense during different growth stages (Figs. 5 and 6, Table S3).

Figure 5 The morphology of C. nankingense at different growth stages.

(A) The seedling period 28 days after sowing. (B) The rapid growth period 44 days after sowing. (C) The stable growth and aging period 60 days after sowing.

Figure 6 Expression pattern analysis of 12 OsNAC7 subfamily genes in different tissues of C. nankingense at different growth stages.

The expression of root at 28d was used as a reference. The x-coordinate represents the different tissues. The y-coordinate represents the expression level. Different color columns indicate different growth periods. Each diagram represents a different gene. Different lower case letters indicate statistically significant differences at the P < 0.05 level. (A–L) The letters represent the different genes in the OsNAC7 subfamily.

As a results, all the 12 members of the C. nankingense OsNAC7 subfamily played a role in roots, stems, and leaves. The expression levels of most of the OsNAC7 genes were generally low in the roots and leaves but high in the stems. However, differences in the expression levels were observed during different root, stem, and leaf developmental stages. Ten out of the 12 members in the OsNAC7 subfamily were expressed at higher levels in the stems at 44 d, including CHR00068684, CHR00042500, CHR00009966, CHR00043953, CHR00073261, CHR00032377, CHR00026420, CHR00003673, CHR00027311, and CHR00043175. CHR00048081 was expressed at high levels in the stems at 60 d. CHR00020838 was expressed at high levels in the stems during the whole growth process, which were much higher than those in roots and leaves. Five out of the 12 members in the OsNAC7 subfamily were significantly related to root growth. Among the 12 members. CHR00048081-RA was expressed at a higher level at 28 d, CHR00020838-RA and CHR00043175-RA had higher levels at 44 d, and CHR00009966-RA and CHR00073261-RA had higher levels at 44 d and 60 d. Five out of the 12 genes were expressed at higher levels in leaves at 60 d, such as CHR00043953-RA, CHR00032377-RA, CHR00026420-RA, CHR00003673-RA, and CHR00043175-RA.

Expression analysis of the C. nankingense OsNAC7 subfamily genes under the osmotic and salt stress treatments

All A. thaliana OsNAC7 subfamily genes are associated with secondary metabolism (Yamaguchi et al., 2008). However, among the O. sativa OsNAC7 subfamily genes, ONAC106 inhibits leaf senescence and also increases salt damage and tillering angle (Fang et al., 2008; Sakuraba et al., 2015); the expression levels of ONAC052, ONAC056, and ONAC084 are downregulated under high salt and drought conditions (Sun et al., 2015). Here, we analyzed the expression levels of the C. nankingense OsNAC7 subfamily genes under osmotic and salt stresses (Fig. 7, Fig. 8, Table S3).

Figure 7 Expression pattern analysis of 12 OsNAC7 subfamily genes in C. nankingense under different treatment times of osmotic stress induced by 20% PEG6000.

The expression level of 0h was used as a reference. The x-coordinate represents the processing time. The y-coordinate represents the expression level. Each diagram represents a different gene. Different lower case letters indicate statistically significant differences at the P < 0.05 level. (A–L) The letters represent the different genes in the OsNAC7 subfamily.

Figure 8 Expression pattern analysis of 12 OsNAC7 subfamily genes in C. nankingense under different treatment times of salt stress induced by 200 mM NaCl.

The expression level of 0h was used as a reference. The x-coordinate represents the processing time. The y-coordinate represents the expression level. Each diagram represents a different gene. Different lower case letters indicate statistically significant differences at the P < 0.05 level. (A–L) The letters represent the different genes in the OsNAC7 subfamily.

The results of gene expression under osmotic stress (Fig. 7) revealed that all of the 12 genes in the OsNAC7 subfamily were regulated by osmotic stress, but their expression patterns were not consistent. Among them, the expression levels of CHR00003673-RA, CHR00027311-RA, and CHR00043175-RA continued to increase after treatment, indicating that these three genes may play a key role improving the resistance to osmotic stress in C. nankingense. The CHR00026420-RA expression fluctuated but was always higher than that of the control group, indicating that this gene is expressed during osmotic stress. CHR00069684-RA showed an obvious response to osmotic stress. The expression of CHR00009966-RA and CHR00048081-RA was inhibited at the initial stage of stress, and then gradually returned to normal expression levels. The response of CHR00042500-RA and CHR00020838-RA was not obvious, and CHR00043953-RA and CHR00073261-RA were severely inhibited by osmotic stress.

The gene expression results under salt stress (Fig. 8) revealed that all of the 12 genes in the OsNAC7 subfamily responded to salt stress, and were initially upregulated and then downregulated or were first downregulated and then upregulated; all decreased to the expression level of those of the control group or below by 36 h. The expression levels of CHR00069684-RA, CHR00009966-RA, CHR00048081-RA, CHR00043953-RA, CHR00026420-RA, CHR00003673-RA, CHR00027311-RA, and CHR00043175-RA increased first at different time points and then decreased. The significantly upregulated expression levels of these eight genes indicate that they may be important in the salt stress response. CHR00042500-RA, CHR00020838-RA, CHR00073261-RA, and CHR00032377-RA were slightly inhibited by salt stress, although their initial responses to salt stress were not obvious.

Subcellular localization and functional analysis

In this study, CHR00069684 that was closely related to the NST protein was used to construct the super35S::CHR00069684::GFP vector, which was then transformed into large leaf tobacco for subcellular localization observation and functional validation. GFP signals were observed in the nucleus and cell membrane of leaves, stems, and roots of super35S::GFP transformed tobacco plants (Fig. 9). No fluorescence signal was detected in the super35S::CHR00069684::GFP transformed tobacco leaves (Fig. S3), whereas there were strong GFP signals in the cell membrane of their roots and stems (Figs. 9E and 9F). The reason why the 35S promoter-guided CHR00069684 expression showed tissue specificity need to be further studied in the future.

Figure 9 Subcellular localization of transgenic tobacco.

(A–D) Subcellular localization of super35S::GFP protein in root (A), leaf (B), root tip (C), and stem (D). (a–d) Enlarged image of GFP in root (a), leaf (b), root tip (c), and stem (d). (E, F) Subcellular localization of super35S::CHR00069684::GFP fusion protein in stem (E) and root (F). (e, f) Enlarged image of CHR00069684::GFP fusion protein in stem (e) and root (f). The red box area is the enlarged area.

The super35S::CHR00069684::GFP transformed and control tobacco plants were treated with 200 mM NaCl, 200 mM ABA, or a low temperature of 4 °C (Fig. 10). The results showed that the transgenic tobacco plants were more resistant to salt and low temperature and more sensitive to ABA than the control plants. Moreover, the differences in the growth and development of tobacco plants were observed (Fig. 11); the flowering time of the transgenic tobacco was 10 d earlier than that of the control plants.

Figure 10 The growth status of tobacco seedlings treated with 200 mmol/L NaCl stress, 4 °C low temperature stress and 200 µmol/L ABA treatment.

(A–C) Growth of super35S::CHR00069684::GFP transgenic tobacco seedlings under NaCl (A), ABA (B) and 4 °C (C). (D–F) Growth of super35S::GFP transgenic tobacco seedlings under NaCl (D), ABA (E) and 4 °C (F). All treatments were grown in stress medium for 30 days.

Figure 11 Growth and development of transgenic tobacco at 43 days.

(A) The growth and development status of super35S::CHR00069684::GFP transgenic tobacco. (B) The growth and development status of super35S:: GFP transgenic tobacco. Both groups of tobacco grew in the same environment and underwent the same conservation management.

Discussion

According to the phylogenetic tree (Fig. 1) and motif analyses (Fig. 4) of CnNAC TFs, we found that the distinct NAC TF subfamilies each containing the most closely related members exhibited highly similar motif components. This included members in the OsNAC7 subfamily (Fig. 12), all of which possess motifs 1–6. The members clustering with ANAC012, ANAC066, ANAC043 (Zhong, Demura & Ye, 2006; Mitsuda et al., 2007; Zhong & Ye, 2015), ANAC033, ANAC070, and ANAC015 (Fendrych et al., 2014; Bennett et al., 2010), which were annotated as NST1–NST3, URP7, and BRN1/2, respectively, have the same motifs 1–7, and the remaining part was clustered with VND1–VND7 (Zhou, Zhong & Ye, 2014; Endo, Yamaguchi & Tamura, 2015; Soyano et al., 2008; Yamaguchi et al., 2010), which lack the motif 6 or motif 7 (Fig. 4). These results indicate that the CnNAC TFs in the same subfamily have similar motifs and similar functions (Zhong, Demura & Ye, 2006; Mitsuda et al., 2007; Zhong & Ye, 2015; Fendrych et al., 2014; Bennett et al., 2010; Zhou, Zhong & Ye, 2014; Endo, Yamaguchi & Tamura, 2015; Soyano et al., 2008; Yamaguchi et al., 2010).

Figure 12 Sequence alignment and motif analysis of OsNAC7 subfamily genes.

Different color frames represent different motif regions. Different colored boxes show the positions of different motifs in genes.

The NAC TFs are involved in various stages of plant growth and development and the responses to various stressors (Yang et al., 2019). It was demonstrated in O. sativa, A. thaliana, and Asteraceae (Table 2) that most NAC TFs regulate plant growth, secondary metabolism, and growth and development time by changing root or stem growth, and further regulate the resistance of plants to biotic and abiotic stressors, but each subfamily had specific characteristics (Yang et al., 2019). Compared to studies of Chrysanthemum and O. sativa, the ATAF subfamily member AaNAC1 (NAC1 in A. annua, KX082975.1) (Yang et al., 2016), which was similar to CHR00002022-RA, CHR00045168-RA, and DgNAC1 (NAC1 in D. grandiform) (Liu et al., 2011), has the same function as the O. sativa ATAF subfamily genes OsNAC5 and OsNAC6 (Nakashima et al., 2007; Jeong et al., 2013; Rachmat et al., 2014; Chung et al., 2009). All of these genes improve drought resistance, salt tolerance, and low temperature tolerance of plants. The ONAC022 subfamily member DlNAC1 (NAC1 in D. nankingense) (Yang et al., 2016), which was similar to CHR00056910-RA and OsNAC063 (Yang et al., 2016; Yokotani et al., 2009; Hong et al., 2016), is involved in regulating drought-tolerance and salt-tolerance in plants. The functions of members belonging to the ATAF and ONAC022 subfamilies in chrysanthemum and rice are essentially the same, suggesting that closely related genes often have similar functions, so analyses of phylogenetic trees can be used to predict the functions of genes in the same subfamily. Through the NAC gene functions of O. sativa and A. thaliana, it has been speculated that the CnNAC genes in the NAC1 subfamily may improve the overall stress resistance of plants (Wang et al., 2018). The NAM subfamily genes improve tolerance to drought and high salt stress, which affect plant height and flowering time, and accelerate leaf senescence (Mao et al., 2017; Shen et al., 2017; Chen et al., 2015; Zheng et al., 2009). The NAP subfamily genes enhance drought tolerance and salt tolerance in plants by influencing the size of the epidermis, cortex, and stellate cells in roots (Jeong et al., 2010; Chen et al., 2014). The ONAC001 subfamily genes are induced by salt and drought stresses (Bo et al., 2019; Gao et al., 2009), the TERN subfamily genes may be associated with disease resistance (Sun et al., 2013), and the SENU5 subfamily genes enhance plant tolerance to salt, alkali, and drought stress (Dong et al., 2018; Liu & He, 2019).

Table 2 Research status of NAC family of rice and chrysanthemum.

Gene name	Subfamily	Function annotation	
ONAC029/031	OsNAC7	Through alternative splicing, the thickening of fibrous cell walls during wood formation was regulates (Liu et al., 2020)	
OsNAC106	OsNAC7	A negatively regulated NAC transcription factor. Osnac106 deletion mutants showed enhanced salt tolerance (Sakuraba et al., 2015)	
OsSND2	OsNAC7	The AtSND2 homologous gene. The overexpressed transgenic plants showed leaf curl, increased cellulose content (Kubo et al., 2005)	
OsNAC5/ONAC071/AK102475	ATAF	It can enhance the drought resistance, salt tolerance and cold tolerance of plants, and the phenotype is not affected (Jeong et al., 2013)	
OsNAC6/ONAC048/AK068392	ATAF	The homologous gene of OsNAC5. The overexpression of OsNAC6 enhanced the tolerance of plants to high salt, drought and cold and its overexpression can also reduce the range of disease and improve the resistance of rice blast (Song et al., 2011; Rachmat et al., 2014; Chung et al., 2009)	
OsNAC60	NAC1	OsNAC60 overexpression enhanced the resistance of transgenic plants to rice blast (Wang et al., 2018)	
OsNAC2/ONAC004/AK061745	NAM	Negatively regulated NAC TF. It can affect plant height and flowering time, reduce chlorophyll level, and accelerate leaf senescence (Mao et al., 2017; Shen et al., 2017; Chen et al., 2015)	
ONAC45	NAM	Under drought and high salt stress, the survival rate of plants with overexpression of this gene was higher than that of wild-type (Zheng et al., 2009)	
OsNAC10	NAP	The root epidermis, cortex and column cells of overexpressed rice plants increased significantly, and the water absorption capacity was enhanced, which enhanced the drought and salt tolerance of the plants (Zheng et al., 2009)	
OsNAC041	ONAC001	Induced expression under salt stress. The rice Osnac041 mutant had higher plant height and stronger salt stress sensitivity (Bo et al., 2019)	
OsNAC52	ONAC001	In A. thaliana overexpressed plants, transpiration of leaves decreased and drought resistance increased (Gao et al., 2009)	
ONAC063	ONAC022	Improved salt tolerance of transgenic A. thaliana. Its overexpression related to salt stress were up-regulated (Yokotani et al., 2009)	
OsNAC022	ONAC022	In overexpressed plants, the water loss rate and transpiration rate decreased, and drought and salt tolerance were enhanced (Hong et al., 2016)	
OsNAC122/OsNAC131	TERN	These two are homologous genes. In rice plants silenced by VIGS method, the ability to infect rice blast was enhanced (Sun et al., 2013)	
OsNAP	NAP	Its overexpression could enhance the content of stress related genes, and improve the resistance of rice to different stress (Chen et al., 2014)	
AaNAC1	ATAF	Its overexpression can increase the content of artemisinin and improve resistance to drought and botrytis cinerea in A. annua (Ko et al., 2007)	
DlNAC1	ONAC022	It can involve in regulating plant tolerance to drought and salinity, and can improve high temperature tolerance of tobacco (Yang et al., 2016)	
ClNAC9	SENU5	It can enhance the tolerance of transgenic A thaliana to salt, alkali and drought stress (Dong et al., 2018). Its transgenic plants chrysanthemum niU9717 can improve the resistance under salinization and drought stress (Liu & He, 2019)	

Moreover, the two unknown subfamilies in CnNAC TFs (i.e., unclassified 1 and unclassified 2) contained more genes than the other subfamilies and had the most abundant numbers of motifs among the CnNAC TFs, indicating that these two unknown subfamilies may have diverse functions and may be involved in a wider range of growth and stress regulatory functions. The diversity of NAC family gene functions may be the reason why chrysanthemums, as a young and highly evolved large family, have strong adaptability to changing environments.

According to the subcellular localization and transmembrane analysis results (Table S1), most CnNAC TFs were related to nuclear genes; however, most of them were mainly expressed outside the membrane. For example, CHR00069684 was a nuclear gene, but was expressed outside the membrane (Table S1); CHR00069684 was expressed in the cell membrane of tobacco, and could only be observed in the stems and roots of transgenic tobacco plants (Fig. 9). These results are consistent with the finding that TFs found in poplar can be localized to the nucleus and cytoplasm at the same time, but they are transferred to and expressed in the nucleus when regulated by an external stress signal (Liu et al., 2020).

According to the phylogenetic tree (Fig. 1), the C. nankingense OsNAC7 subfamily members, including CHR00069684-RA, CHR00042500-RA, CHR00020838-RA, and CHR00009966-RA, were annotated as NST1–NST3 and have higher expression levels in the C. nankingense stems at 44 d (Fig. 6), which showed similar functions to the NST protein in A. thaliana (Zhong, Demura & Ye, 2006; Mitsuda et al., 2007; Zhong & Ye, 2015; Kubo et al., 2005; Ko et al., 2007). Moreover, CHR00009966-RA likely played a more important role in the root growth. CHR00048081-RA, CHR00043953-RA, and CHR00073261-RA were annotated as URP7, BRN2, and BRN1 respectively, which are mainly responsible for the root cap. The expression of these three genes in the C. nankingense roots was similar to that of the URP7 and BRN1/2 genes in A. thaliana (Fendrych et al., 2014; Bennett et al., 2010). However, the expression of these three genes was higher in stems than in roots, indicating that CHR00048081-RA might play a regulatory role in the root growth and aging of the stem; CHR00043953-RA might play a regulatory role in the rapid growth of the stem and aging of the leaf; and CHR00073261-RA might play a regulatory role in the rapid growth of the stem. CHR00032377-RA, CHR00026420-RA, CHR00003673-RA, CHR00027311-RA, and CHR00043175-RA corresponded to the VND1, VND2, VND3, VND4, VND5, VND6, and VND7 genes, which co-participate in regulating the thickening of the secondary cell walls in A. thaliana (Zhou, Zhong & Ye, 2014; Endo, Yamaguchi & Tamura, 2015; Soyano et al., 2008; Yamaguchi et al., 2010; Kubo et al., 2005). These genes were all involved in the stem growth and the aging process of leaves at the later growth stages (Fig. 6). Besides, CHR00043175-RA was also involved in the root growth.

All genes responded to the osmotic (Fig. 7) and salt stresses (Fig. 8), which has only been studied in the ONAC106 gene of the OsNAC7 subfamily in rice (Fang et al., 2008; Sakuraba et al., 2015). Among them, CHR00026420-RA, CHR00003673-RA, CHR00027311-RA, and CHR00043175-RA may play a key role improving the resistance to osmotic stress in chrysanthemum. CHR00069684-RA, CHR00009966-RA, CHR00048081-RA, CHR00043953-RA, CHR00026420-RA, and CHR00003673-RA may be important genes in the salt stress response. To test this hypothesis, the super35S::CHR00069684::GFP transgenic tobacco lines were treated with salt stress, and the results confirmed the active function of CHR00069684 in the regulation of salt stress. In addition, the results obtained from ABA and low temperature treatments of the super35S::CHR00069684::GFP transgenic tobacco plants indicated that CHR00069684 enhanced ABA sensitivity and low temperature stress resistance in the transgenic tobacco plants, leading to reduced growth potential and premature flowering. Functional analysis on CHR00069684 confirmed the dual roles of the OsNAC7 subfamily genes in the growth and stress regulation of plants. However, the function of the OsNAC7 subfamily members during abiotic stress remains speculative, and many other functions of the OsNAC7 subfamily genes remain to be further explored.

Conclusions

In this study, the CnNAC TFs were divided into two groups with 19 subfamilies according to the phylogenetic tree. These 19 subfamilies consisted of 17 known subfamilies and two unknown subfamilies. The conserved motifs, subcellular localization, transmembrane localization, and physicochemical properties of the CnNAC TFs were comprehensively analyzed. Combined with research on the NAC family genes of A. thaliana, O. sativa, and Asteraceae, the functions of the CnNAC TFs were investigated. Analyses on the expression of the 12 CnNAC genes in the OsNAC7 subfamily under osmotic stress, salt stress, and in different tissues at different time points showed that members of the OsNAC7 subfamily not only played a regulatory role in the growth and development of roots, stems, and leaves of C. nankingense, but also responded to osmotic and salt stresses. These findings may provide new ideas for regulating plant stress resistance and growth. The function of CHR00069684 was verified in transgenic tobacco plants, and it was found that CHR00069684 could confer enhanced resistance to salt and low temperature stresses, improve the ABA sensitivity, and lead to the early flowering of transgenic tobacco. Studies on the CHR00069684 function confirmed the dual roles of the OsNAAC7 subfamily genes in stress regulation and plant growth and development. This study provides a theoretical basis for studying NAC TFs, the stress tolerance mechanism, and the plant growth process.

Supplemental Information

Supplemental Information 1 Sequence information of the super1300 vector.

Click here for additional data file.

Supplemental Information 2 Sequence information of the CHR00069684 gene.

Click here for additional data file.

Supplemental Information 3 The original file of the phylogenetic tree.

Click here for additional data file.

Supplemental Information 4 Analysis of the number of genes contained in different subfamilies.

The x-coordinate shows the name of the subfamily and the y-coordinate shows the number of genes.

Click here for additional data file.

Supplemental Information 5 Statistics of genes containing different motif Numbers.

The x-coordinate represents the serial number of motifs, and the y-coordinate represents the number of genes.

Click here for additional data file.

Supplemental Information 6 Subcellular localization of CHR00069684::GFP fusion protein in transgenic tobacco leaves.

Click here for additional data file.

Supplemental Information 7 Results of subcellular localization and transmembrane analysis.

Click here for additional data file.

Supplemental Information 8 Results of physicochemical properties analysis.

Click here for additional data file.

Supplemental Information 9 Raw data of Figure 6, Figure 7 and Figure 8.

Click here for additional data file.

Additional Information and Declarations

Competing Interests

Author Contributions

Data Availability

The authors declare that they have no competing interests.

Hai Wang conceived and designed the experiments, performed the experiments, analyzed the data, prepared figures and/or tables, authored or reviewed drafts of the paper, and approved the final draft.

Tong Li performed the experiments, analyzed the data, prepared figures and/or tables, and approved the final draft.

Wei Li performed the experiments, analyzed the data, prepared figures and/or tables, and approved the final draft.

Wang Wang performed the experiments, analyzed the data, prepared figures and/or tables, and approved the final draft.

Huien Zhao conceived and designed the experiments, performed the experiments, analyzed the data, prepared figures and/or tables, authored or reviewed drafts of the paper, and approved the final draft.

The following information was supplied regarding data availability:

Sequence data are available at the Chrysanthemum Genome Database: http://www.amwayabrc.com/index.html. The raw experimental data are available as Supplemental Files.

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
