# Peer review of "Identification and analysis of Chrysanthemum nankingense NAC transcription factors and an expression analysis of OsNAC7 subfamily members"

_PeerJ, doi:10.7717/peerj.11505_

## Round 0.1 · original submission · Major Revisions

All three experts thought that your work was interesting. However, they all brought up numerous good suggestions for your manuscript.

Reviewer 1 ·

Basic reporting

The general structure of the artical are intact and clear. Introduction is straight and narrow. The main text is generally easy to read, however, almost all the figures lack figure legend to aim reading. The data roughly support the main theme.

Experimental design

The experimental design is rational. Most of them are descriptive. However, more web-lab experiments are needed to support the linkage of CnNACs and stresses.

Validity of the findings

The descriptive part is fine. More web-lab experiments are needed to support the linkage of CnNACs and stresses.

Additional comments

In this manuscript by Wang et al., the NAC transcription factors of Chrysanthemum nankingense are systematically analyzed by bioinformatics tools. Those features of the CnNACs including phylogenetic relationship, conserved peptide motifs, putative subcellular localization, and rough expression patterns are summarized. In addition, some individual genes are selected to analyze the potential stress response. As mentioned by the authors, the Chrysanthemum nankingenseare shows a good performance of stress resistance; meanwhile, the NAC transcription factors are deeply involved in the stress response in many plant species. This is a good selling point to emphasize the significance of this work, while the support is relatively weak. A general description of CnNACs is good but not sufficient to profile the link between CnNACs and stresses. Since the functional analysis of several CnNACs might be hard to investigate due to lacking reliable transgenic system, ectopic while in planta analysis should be conducted at least in Arabidopsis. For example, based on the stress responses showed herein, some candidates can be transformed into Arabidopsis to see the in planta effects.
Some detailed comments are as following:
1. The manuscript has two title pages. However, the affiliations are inconsistent.
2. Although I can understand the figures, legends are always required for any figures.
3. The reference citations are not well-formatted in the text. For example, in line 38, the “[4]” should be superscript as other citations. Please carefully check and fix those errors one by one.
4. Line 65, ABA is not abiotic stress, though it is important for the response of abiotic stress. Line 66, the virus resistance is related to biotic stress rather than abiotic stress. The sentence in the line 64-66 is too wordy with redundant information.
5.The third paragraph in the introduction section (line 64-81) is describing the link between NACs and stresses. The sentence in the line 67-68 is not suitable to be put here.
6. In the methods & materials section, it is usually not required to cite figures. Please refer to line 133.
7. The scale bar of Fig.1 is strange. Since the authors do not change the pot for the plant during the growing, the scale bar for the middle and right panel could be mispresented. Please carefully check the raw data.
8. For Fig.2, the bootstrap of the tree is always required. At the lowest level, only the node with the bootstrap higher than 50% is reliable and could be presented. In addition, the multi-alignment data should be provided accordingly.
9. Also, there might be something wrong with the “Rootless phylogenetic trees” as shown in the ANAC063” clade. A root appears there. Please check the raw data.
10. Since the authors have involved rice NACs in the tree, why only 6 rice NACs are analyzed? It should be unbiased to generate the tree, therefore, if the authors want to involve rice NACs, all the rice NACs should be put in the tree. So do the Arabidopsis NACs.
11. More information are needed to support why c3ulxA and d1ut7a are proper templates for modeling.
12. The individual panels in Fig.3 are not easy to read details. The authors may consider arranging the data as a multi-panel figure.
13. Only the prediction of subcellular localization is presented. Since the NAC proteins are transcription factors, it is easy to understand their nuclear localization. However, extracellular localization and membrane localization are also predicted. It is interesting, while the subcellular localization of several typical NAC candidates should be validated by wet-lab experiments. At least, transient expression analysis should be used.
14. Statistical analysis should be applied for Fig. 6,7, and 8.
15. In the discussion section, the authors should not present results in detail.

Reviewer 2 ·

Basic reporting

no comment

Experimental design

no comment

Validity of the findings

no comment

Additional comments

The manuscript by Hai et al. uses bioinformatic, genetic, and molecular approaches to identify NAC transcription factors and determine expression levels of OsNAC7 subfamily members in C. nankingense. The authors have generated a solid amount of data and the conclusions are justified by the results. Besides that, the details in some experiments are missing or need to be clarified in the text and legends.

Line 119: italic “A. thaliana” and “O. sativa”.

Figure 2: It would be better to differentiate which are from A. thaliana, O. sativa, and C. nankingense in legends.

Figure 4: brief explain the meaning of E-value, sites, and width.

Figure 6: Explain relative expression to what? And add statistical analysis.

Figure 7 and 8: include the meaning 20% PEG6000 and 200 mM NaCl for osmotic and salt stress, respectively.

Supplementary data of Figure6_7_8 is not cited in the manuscript.

Reviewer 3 ·

Basic reporting

Abbreviations are not properly defined. For example, NAM, ATAF, and CUC (line 36-37); CnNAC (line 98)

Statements lack citations. For example, line 69-71; line 92-93; line 260; line 298-303

Figure 1 does not provide any meaningful information.

Experimental design

The authors should state clearly what is used as the control sample for RT-qPCR experiments related to Figure 6-8.

Validity of the findings

Change in gene expression during stress treatment does not necessarily mean the genes are important regulators of stress response. The authors should not over interpret the results.

---

## Round 0.2 · Minor Revisions

Two experts still have some suggestions to improve your manuscript. Please revise the manuscript accordingly.

Reviewer 1 ·

Basic reporting

NIL.

Experimental design

NIL.

Validity of the findings

NIL.

Additional comments

I appreciate the efforts of the authors to improve the manuscript. The previous concerns are addressed accordingly. The statistical analysis had been added in Fig. 6,7, and 8, but they need to be described in the legends.
However, the revised text, including that in the main text and methods, MUST be edited by a native speaker. A lot of grammatical errors are there, and the sentences are not readable. In addition, the presentation of the legends is really bad, especially for the multi-panel figures. A key point is that, NEVER to use “A”, “B”, “C”, etc as a subject in a descriptive sentence. “A”, “B”, “C”, etc, are only the labels of the panels.
For example:
>> “A means 28 days after sowing, which is the seedling stage.”
--- “A” does not mean that at all.
Do revise it like that, “A. The picture showing the seedling 28 days after sowing.”

>> “A, B, C, D is a subcellular observation of 1300: : GFP. A is the root of 1300::GFP, B is the leaf of 1300::GFP, C is the root tip of 1300::GFP, D is the stem of 1300::GFP.”
---“A” is not the observation, thus the whole sentence is nonsense.
Do revise it like that, “A-D. Subcellular localization of GFP in root (A), leaf (B), root tip (C), and stem (D).”

I can’t point them one by one, but DO please revise all such sentences entirely.

Reviewer 2 ·

Basic reporting

The structure of revised manuscript is clear with all the required sections.

Experimental design

The experimental design is well defined to answer questions.

Validity of the findings

The findings are well described and discussed.

Additional comments

The revised manuscript addresses the reviewers' comments with detailed descriptions and additional analyses/experiments. I do not have further comments.

Reviewer 3 ·

Basic reporting

The English in the present manuscript is not of publication quality and requires major improvement.

The figure legends are particularly confusing. I highly recommend the authors to follow standard figure legends formats of scientific papers. Also, in Figure 2, there are no individual descriptions of the three panels; in Figure 6 and Figure 7, there is no description of what the small letters (a, b, and c) above the bars represent.

Experimental design

It's nice to add the CHR00069684::GFP experiment. However, important details of this experiment are missing. For example:

1) information and references of pSuper1300

2) DNA sequence or the genomic coordinates of the ORF of CHR00069684 inserted into the vector.

3) promoter driving the expression of CHR00069684::GFP or GFP

4) meaning of CHR00069684::GFP. Is it a fusion protein?

Validity of the findings

Since only the ORF of CHR00069684 was cloned into pSuper1300 vector, I'm assuming the vector contains some ubiquitously expressed promoters to drive transgene expression. If so, why is pSuper1300::GFP expressed in leaves, while pSuper1300::CHR00069684::GFP is NOT?

I cannot tell whether the fluorescence is in cytoplasm or on cell membrane in Figure 9. The authors should provide images with higher magnifications and better qualities. Staining the tissues with cell membrane, cytoplasm, and nucleus markers would also help. Also, the authors should show images of the leaves of the pSuper1300::CHR00069684::GFP line to support their claim that there is no expression of CHR00069684::GFP in leaves.

---

## Round 0.3 · accepted · Accept

The manuscript has been improved after revisions.